# Posttranscriptional Regulation in Response to Different Environmental Stresses in *Campylobacter jejuni*

Stephen Li,[a] Jenna Lam,[a] Leonidas Souliotis,[a] ⬤Mohammad Tauqeer Alam,[b] ⬤Chrystala Constantinidou[a]

[a]Warwick Medical School, University of Warwick, Coventry, United Kingdom
[b]Department of Biology, College of Science, United Arab Emirates University, Al-Ain, United Arab Emirates

Stephen Li and Jenna Lam contributed equally to this article. Author order was determined by the corresponding author after negotiation.

**ABSTRACT** The survival strategies that *Campylobacter jejuni* (*C. jejuni*) employ throughout its transmission and infection life cycles remain largely elusive. Specifically, there is a lack of understanding about the posttranscriptional regulation of stress adaptations resulting from small noncoding RNAs (sRNAs). Published *C. jejuni* sRNAs have been discovered in specific conditions but with limited insights into their biological activities. Many more sRNAs are yet to be discovered as they may be condition-dependent. Here, we have generated transcriptomic data from 21 host- and transmission-relevant conditions. The data uncovered transcription start sites, expression patterns and posttranscriptional regulation during various stress conditions. This data set helped predict a list of putative sRNAs. We further explored the sRNAs' biological functions by integrating differential gene expression analysis, coexpression analysis, and genome-wide sRNA target prediction. The results showed that the *C. jejuni* gene expression was influenced primarily by nutrient deprivation and food storage conditions. Further exploration revealed a putative sRNA (CjSA21) that targeted *tlp1* to *4* under food processing conditions. *tlp1* to *4* are transcripts that encode methyl-accepting chemotaxis proteins (MCPs), which are responsible for chemosensing. These results suggested CjSA21 inhibits chemotaxis and promotes survival under food processing conditions. This study presents the broader research community with a comprehensive data set and highlights a novel sRNA as a potential chemotaxis inhibitor.

**IMPORTANCE** The foodborne pathogen *C. jejuni* is a significant challenge for the global health care system. It is crucial to investigate *C. jejuni* posttranscriptional regulation by small RNAs (sRNAs) in order to understand how it adapts to different stress conditions. However, limited data are available for investigating sRNA activity under stress. In this study, we generate gene expression data of *C. jejuni* under 21 stress conditions. Our data analysis indicates that one of the novel sRNAs mediates the adaptation to food processing conditions. Results from our work shed light on the posttranscriptional regulation of *C. jejuni* and identify an sRNA associated with food safety.

**KEYWORDS** *C. jejuni*, bioinformatics, sRNA, signal transduction, transcriptional regulation

**C**ampylobacter jejuni (*C. jejuni*) is a leading foodborne pathogen that infects the gastrointestinal (GI) tract and causes inflammation, abdominal pain, fever, and diarrhea (1–4). Occasionally, *Campylobacter* infection campylobacteriosis can lead to more severe complications such as reactive arthritis (RA), Guillain-Barre syndrome (GBS) and long-term childhood physical and cognitive impairments (5–7).

While *C. jejuni* is a fastidious organism to culture in the laboratory, its ability to adapt to stress during transmission makes it a successful pathogen. *C. jejuni* experiences environmental

**Ad Hoc Peer Reviewer** ⬤ Victor DiRita, Michigan State University

Address correspondence to Mohammad Tauqeer Alam, mtalam@uaeu.ac.ae, or Chrystala Constantinidou, C.I.Constantinidou@warwick.ac.uk.

The authors declare no conflict of interest.

stresses such as but not limited to bile salt, temperature variations, reactive oxygen species (ROS), and host iron limitation (8–10). It remains unclear how *C. jejuni* adapts to various environmental stresses with such a small genome (1.6 Mb) that carries only three annotated sigma factors and no conserved global stress response regulators like *rpoS* found in other Gram-negative bacteria (11).

Emerging evidence has brought to light the importance of sRNAs in stress response, infection, and antibiotic resistance (12). sRNAs are short molecules ranging from 50 to 500 nucleotides that regulate biological processes by base-pairing with specific or multiple mRNA targets (13–15). sRNAs can inhibit the translation of mRNA targets by physically blocking the ribosomal binding site (RBS) (16, 17) or facilitating RNase E degradation (18–20). sRNA-mRNA interactions can also activate translation initiation by disrupting secondary structures to expose the RBS (21–23) or enhancing mRNA stability through inhibiting RNase E digestion (24, 25). Examples of sRNA-mediated pathways include the tricarboxylic acid (TCA) cycle, amino acid uptake, oxidative stress response, iron homeostasis, virulence, and antibiotic resistance (26–34).

The lack of identified global RNA-binding proteins has hindered the discovery of *C. jejuni* sRNAs. There have been several published transcriptomic and RNA-Seq data sets (35–39) that have enabled the detection of potential novel sRNAs in *C. jejuni*. Of these, references Dugar et al. (35) and Porcelli et al. (36) focused on standard growth conditions, while Butcher and Stintzi (37) investigated iron homeostasis, and Handley et al. (39) looked at inactivation of Fur and PerR. Notably, Taveirne et al. (38) identified sRNA expression from an *in vivo* chicken model, although this was conducted in *C. jejuni* strain DRH212, which is a streptomycin resistant derivative of strain 81-176 rather than the more widely studied NCTC 11186 used in the aforementioned data sets. These studies have provided a beneficial insight into the transcriptional landscape of *C. jejuni*. However, they are not necessarily comparable due to differences in growth and stress conditions and may have missed condition-specific sRNAs outside the scope of their study. Bacterial sRNAs play a critical role in regulating stress responses as reviewed in (40); therefore, a comprehensive approach is needed to uncover the untapped reserve of potential condition-dependent novel sRNAs in order to fully understand the complex nature of *C. jejuni* posttranscriptional regulation. Among experimentally confirmed sRNAs, most of their condition-specific activities remain largely unknown.

This study aims to determine the *C. jejuni* transcriptional landscape across 21 host- and transmission-relevant conditions. This involves enriching transcriptional start sites (TSS) from all conditions with Streptavidin enrichment using the Cappable-seq protocol (41). Moreover, expression coverage from all conditions was obtained using the RNAtag-Seq protocol, which involves barcoding all samples before pooling them into one tube for simultaneous cDNA library preparation (42).

To illustrate the potential of studying condition-specific sRNA activities from our data sets, the RNAtag-Seq coverage and TSS data in this study, alongside several published data sets, were used to derive a list of potential sRNA candidates. The RNAtag-Seq data were further analyzed to construct a global sRNA-mRNA interaction network. Among the putative sRNAs, one appeared to inhibit mRNAs encoding transducer-like proteins (Tlps) and may inhibit chemotaxis, contributing to survival under food processing conditions. We have demonstrated how our data set can contribute to understanding *C. jejuni* posttranscriptional regulation.

## RESULTS

**TSS identification by Cappable-seq.** Previous studies by Dugar et al. (35) and Porcelli et al. (36) have found TSS in *C. jejuni* NCTC11168 wild-type strain under standard laboratory growth conditions and Handley et al. (38) in *C. jejuni* NCTC11168 *fur perR* mutant. These studies all used the dRNA-seq approach developed by Sharma et al. (43). Dugar et al. (35) identified 1,905 TSS, with 1,837 TSS retained after clustering TSS less than 10 bp apart. Porcelli et al. (36) and Handley et al. reported 992 and 14 TSS, respectively.

Cappable-seq enriched TSS identified from this study were filtered against the control library, which omitted streptavadin enrichment. Subsequent postprocessing identified

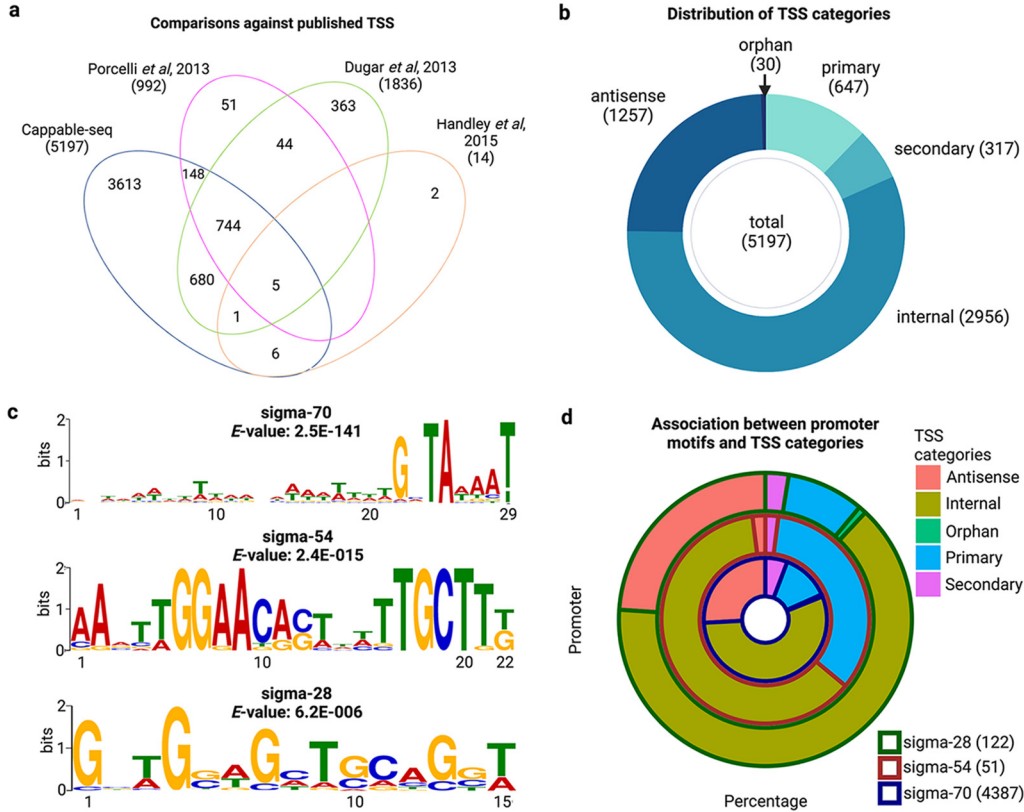

**FIG 1** (a) Comparison of TSS identified from Cappable-seq against TSS identified from published work. The number in the bracket represents the total TSS from each data source. (b) The number of each TSS type. The numbers in the bracket indicate the frequency of each TSS category. (c) MEME logo of sigma-70, sigma-54, and sigma-28 promoter motifs, respectively. The *x* axis displays the nucleotide position of the consensus motif. (d) The distribution TSS categories and promoter motifs. The number in the bracket indicates the number of promoter motifs. Only statistically enriched promoter motifs are shown here.

5197 TSS, of which 3613 TSS were novel (Fig. 1a). TSS 500 bp upstream of the 5′-end of an annotated gene were classified as either primary or secondary TSS. If multiple TSS upstream of the same annotated gene were observed, the most enriched TSS (with the highest RRS) was classified as the primary TSS. In addition, TSS were categorized according to Fig. S3. More than half (2,956 out of 5,197) were internal TSS (Fig. 1b). Moreover, 2,996 novel TSS were statistically enriched and associated with a promoter motif, with the majority of them (1,949 out of 2,996) being internal TSS. In addition, 123 and 143 of these novel TSS are primary and secondary TSS, respectively (Table S1).

The promoter motifs of the 5,197 sequences were identified by submitting the upstream sequences to Multiple Em for Motif Elicitation (MEME)-suite. The search identified the three known sigma factor motifs (Fig. 1c). Both sigma-70 and sigma-54 motifs resembled previously reported motifs (35, 36), with sigma-70 showing an additional strong G upstream. However, sigma-28 showed no clear motif.

The frequency of sigma-70 and sigma-54 motifs among Cappable-seq TSS was calculated by searching with the consensus motif sequences 5′-GNTANAAT and 5′-GG-N9-TCGT that we have identified, respectively, along with the published sigma-28 motif 5′-CGATWT and distance parameters set by Porcelli et al. (36). Two mismatches were allowed for the sigma-70 motif by following the criteria for published promoters. Of all promoter sequences upstream of identified Cappable-seq TSS, 4387 carried a sigma-70 motif, 51 contained a sigma-54 motif and 122 had a sigma-28 motif as visually presented in Fig. 1d. Over half of all promoters for each sigma factor were internal and about a quarter of sigma-70 and sigma-28 promoters were antisense (Fig. 1d). Interestingly, sigma-54 seems to have a higher proportion of primary promoters (33.33%) compared with the other sigma factors.

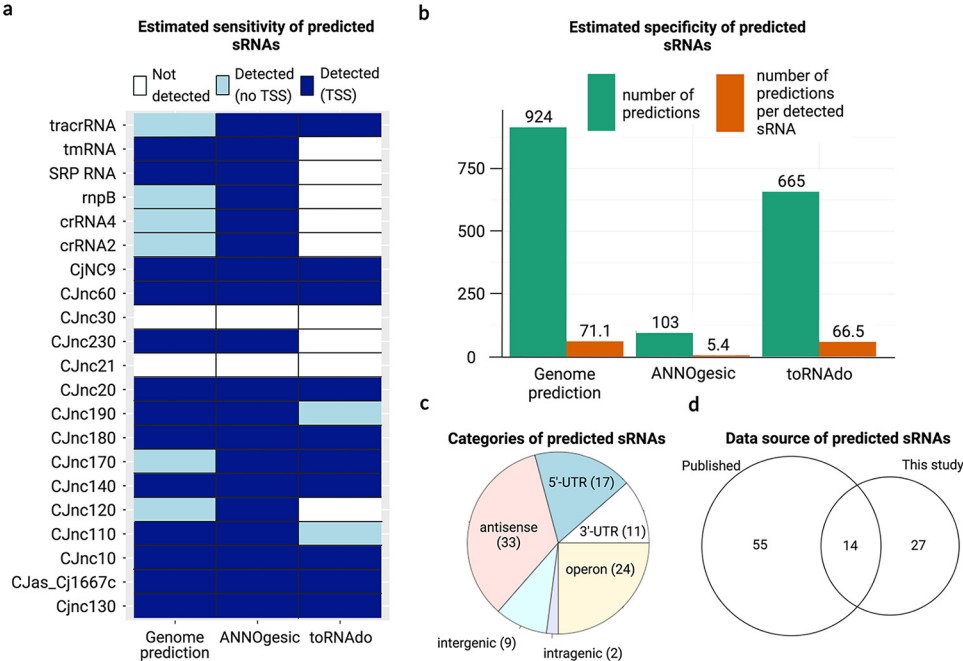

**FIG 2** (a) The detection of published sRNAs (validated by Northern blotting) by three different sRNA approaches. Some published sRNAs were detected by predictions without any upstream TSS, while some predictions consisted of at least one TSS. (b) The total number of TSS-coupled predictions made by each tool and the number of TSS-coupled predictions per detected benchmark sRNA. (c) Distribution of different categories of putative sRNAs. Predictions within 100 bp away from the 5′-ends or 3′-ends of another gene were categorized as UTR-derived sRNAs (5′-UTR or 3′-UTR). Predictions derived from both 5′-UTR and 3′- UTR were labeled as "operon." Any predictions on the opposite strand of another gene(s) were annotated as antisense sRNAs. Predicted sRNAs completely embedded inside another gene (100 bp within both ends) were considered intragenic. Otherwise, any sRNAs with no genes in proximity or the opposite were annotated as intergenic. (d) The distribution of ANNOgesic outputs from different data sources, from data sets published in this study, previous publications, or both.

In order to explore the difference of TSS expression among all Cappable-seq TSS, the TSS expression coverage was calculated in transcripts per million (TPM) with hierarchical clustering of TSS categories and sigma motifs (44) (Fig. S4). The analysis revealed the most highly and widely expressed TSS across all conditions tend to be internal, likely due to the transcription signal from the protein-coding gene where the internal TSS resides. In addition, most antisense TSS have low expression in the majority of the conditions. However, condition-specific antisense TSS expression was observed, especially in the hyperosmotic stress (NaCl) condition. However, there was no clear expression variation observed within the TSS categories and no clustering of TSS with sigma motifs (Fig. S4). Interestingly, most of the sigma-28 antisense TSS were on the opposite strand of metabolic pathways other than flagella-associated genes. This suggests that sigma-28 may regulate metabolic pathways other than flagellar assembly.

**sRNA prediction.** The novel TSS discovered by Cappable-seq may indicate the presence of novel sRNAs. In order to uncover more sRNAs, a list of sRNAs were predicted from several *in silico* tools (see Materials and Method section). sRNA pre-dictions from ANNOgesic (45), toRNAdo (46), and Genome prediction were assessed using 21 northern-blot validated sRNAs with well-defined boundaries from previous studies (35, 36). The key selection criteria were the estimated sensitivity (the number of benchmark sRNAs detected), followed by the estimated specificity (the number of predictions per detected benchmark sRNAs). ANNOgesic detected 19 out of the 21 benchmark sRNAs, with all of them coupled to at least one TSS within 500 bp upstream of the 5′-end (Table S2). While Genome prediction also detected 19 benchmark sRNAs, six of them did not have a TSS near the 5′-end. Meanwhile, toRNAdo only detected 12 benchmark sRNAs (Fig. 2a). ANNOgesic overall predicted 116 sRNAs, 103 of them were associated with a TSS, whereas Genome prediction and toRNAdo predicted a much larger number

of TSS-coupled sRNA, 924 and 665, respectively (Fig. 2b). The ratio of predicted sRNAs to detected validated sRNAs for ANNOgesic was 5.4, compared with 71.1 and 66.5 for Genome prediction and toRNAdo, respectively (Fig. 2b).

Hence, all 116 sRNAs anticipated by ANNOgesic were carried forward and designated CjSAX, where X is an integer between 1 and 116. For instance, the prediction closest to the origin of replication was named CjSA1.

A closer inspection of the 5′ and 3′ boundaries of the ANNOgesic-predicted sRNAs revealed that those that matched with the benchmark sRNAs had start sites with genomic positions less than 10 nucleotides away to the 5′ ends of the benchmark sRNAs. For the ANNOgesic-predicted sRNAs that were associated with multiple TSS, the TSS furthest upstream was selected as the 5′ end, unless gene expression coverage suggested otherwise. The 3′-ends of the ANNOgesic-predicted sRNAs were further refined using similar parameters as toRNAdo (https://github.com/pavsaz/toRNAdo), which filtered out the 3′-end nucleotides with expression coverage at least 5-fold lower than the maximum expression of their corresponding predictions. Predictions that appeared to share similar genome coordinates with annotated genes were removed manually to minimize false positives from transcription signals of annotated genes. After manual correction and removal of predictions without TSS, 96 putative sRNAs were carried forward for downstream analysis.

The majority of the 96 putative sRNAs were either antisense, inside an operon, or UTR-derived. Only nine out of 96 sRNAs were intergenic (Fig. 2c). Further comparison against published sRNAs (including those that were not validated by Northern blotting) showed that 65 of our predicted sRNAs had not been detected by previous studies (35, 36) (Fig. S5), emphasizing the contribution from the data from nonstandard conditions.

BLASTN search showed that most putative sRNAs were found only in *C. jejuni* strains, while few were conserved among other *Campylobacter* but not *Helicobacter* species (Fig. S6). Interestingly, some sRNAs such as CjSA101 and CjSA50 were not conserved even among other *C. jejuni*, suggesting strain-specific regulatory functions. In contrast, candidates including CjSA21 were highly conserved among *Campylobacter* and *Helicobacter* strains. Such conservation suggests common regulatory roles among Epsilonproteobacteria strains (Fig. S7).

Further analysis of the condition-specific expression of TSS showed that 3,736 Cappable-seq TSS were found in the standard laboratory condition (37 M), while 4,400 were found among the other 20 conditions. Only three TSS were expressed in 37 M but not in any other conditions (Fig. S4). Likewise, among Cappable-seq TSS specifically associated with the 96 predicted sRNAs, 69 and 76 TSS were found in standard laboratory conditions (37 M) and nonstandard conditions, respectively. All 69 TSS expressed in 37 M were also expressed in other conditions. Most predicted sRNAs carried TSS that showed varied levels of expression across nonstandard conditions when compared with the standard 37 M condition (Fig. S8). Interestingly, TSS expression of ANNOgesic-predicted sRNAs, such as CjSA21, CjSA54, and CjSA102, was higher in the hyperosmotic replicates (Fig. S8) suggesting that some sRNAs might have stress regulatory roles. Note that some internal TSS may regulate the expression of antisense or UTR-derived sRNAs if the transcript boundaries of those sRNAs reach the antisense of UTR regions of protein-coding genes.

**Overview of RNAtag-seq data.** Further, detailed investigation of the predicted sRNAs aimed to understand the conditions and pathways in which the sRNAs play important roles in stress adaptation. The biological function of the predicted sRNAs was investigated by identifying their potential binding targets, and assuming that the sRNA-target pairs share highly correlated gene expression patterns across all RNAtag-seq samples and have stable binding energy. An sRNA-target network was then built by analyzing RNAtag-seq data with WGCNA, DESeq2, and IntaRNA, using similar criteria to a comparable study that constructed the RNA-RNA interactome from *Staphylococcus aureus* using RNA-seq data sets (47) (Fig. S9a).

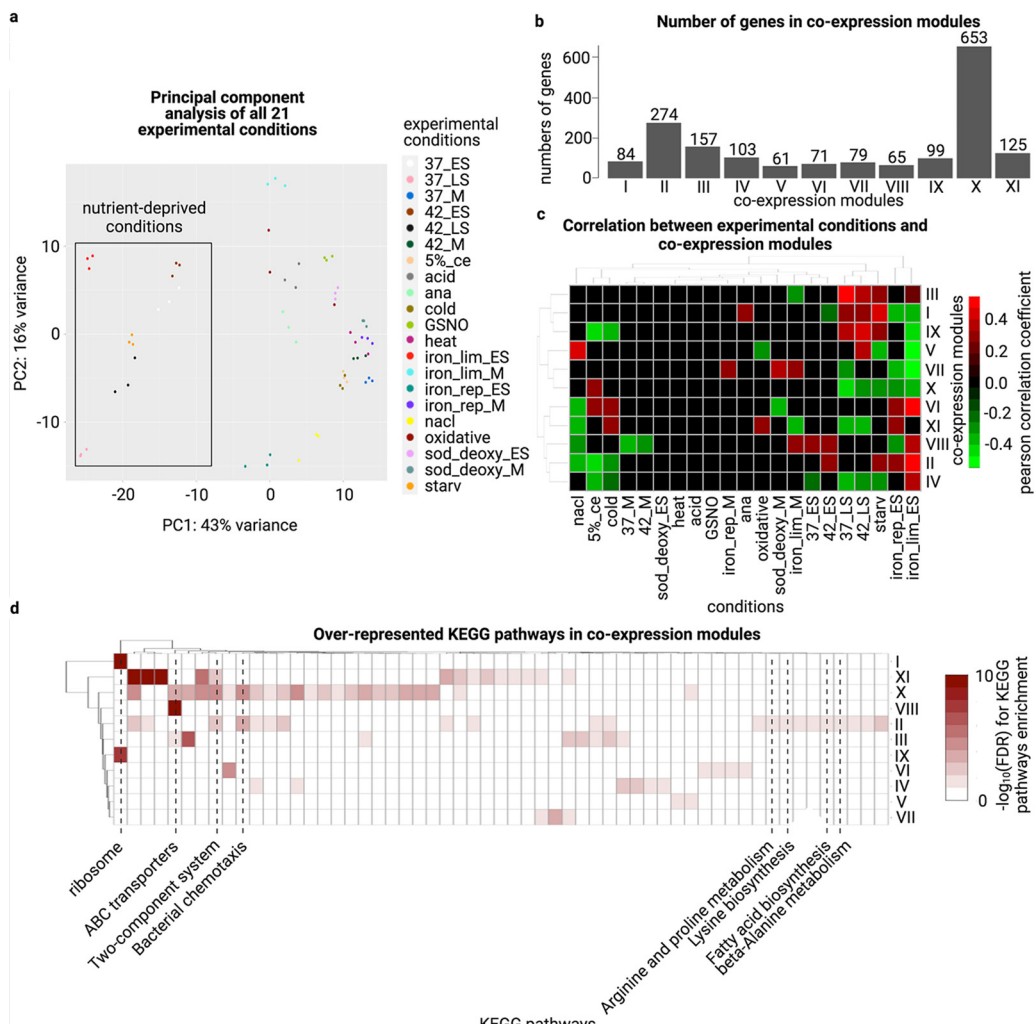

**FIG 3** (a) PCA plot of transformed raw expression data from all 21 conditions. (b) The number of genes assigned to coexpression modules I to XI (c) The Pearson correlation between the eigengene (the first principal component of the gene expression matrix) of each coexpression module and experimental traits. All black boxes indicated a Pearson correlation *P*-value > 0.05. A positive correlation suggested that under the corresponding experimental condition, the overall gene expression in the module increased as well. Likewise, a negative correlation indicated that the overall gene expression of the module decreased under that experimental condition. (d) -log$_{10}$(FDR) of KEGG pathways enrichment for all coexpression modules. Those with FDR > 0.05 are colored in white.

**(i) Coexpression.** Principal component analysis (PCA) and the Euclidean distance visualization of the raw RNAtag-seq expression data showed a distinctive cluster for nutrient-deprived conditions, including the early stationary phase and the late stationary phase in both 37°C and 42°C (37 ES, 37 LS, 42 ES, 42 LS, and iron lim ES), starvation stress (starv) and the early stationary phase under iron limitation (iron lim ES) (Fig. 3a).

Coexpression analysis was conducted on all annotated genes and predicted sRNAs. Based on the gene expression patterns across all RNAtag-seq conditions, we partitioned all genes into 11 coexpression modules using average linkage hierarchical clustering based on their expression patterns and network topologies. The modules were named modules I to XI (Fig. S1d). Module X consisted of 653 genes, more than any other module. In contrast, module V was the smallest module that consisted of only 65 genes (Fig. 3b). Several coexpression modules showed a significant correlation with conditions related to the late stationary phase, the early stationary phase, and starvation. Meanwhile, several stress conditions, including heat stress (heat), acid stress (acid), and nitrosative stress (GSNO), showed no statistically significant correlation (*P*-value 0.05) with any coexpression modules, suggesting these conditions had less impact on

driving coexpression for the 11 coexpression modules (Fig. 3c). Therefore, our WGCNA coexpression analysis suggested that nutrient deprivation was the critical factor for driving gene expression and may involve crosstalks with other stress responses pathways. Expression changes induced by nutrient deprivation were observed among some predicted sRNAs. For example, CjSA110 showed the lowest expression under the late stationary phase, the early stationary phase, and starvation (Fig. S10).

Apart from nutrient deprivation conditions, other stress conditions, including cold stress (cold), 5% chicken exudate (5% ce), and hyperosmotic stress (nacl), also acted as the significant driving force for gene coexpression (Fig. 3c). Some coexpression modules showed enrichment with a small number of KEGG pathways (Fig. 3d). For instance, modules I and IX were only enriched with "Ribosome," while the only enriched pathway of module VIII was "ABC transporters." Meanwhile, modules II, X, and XI consisted of 274, 653, and 125 genes, respectively. They were also enriched with over 10 KEGG pathways. Interestingly, module II was enriched with genes related to signal transduction ("two-component system"), cell motility ("bacterial chemotaxis"), and metabolites biosynthesis ("beta-alanine metabolism," "lysine biosynthesis," "arginine and proline metabolism," "fatty acid biosynthesis"). Hence, module II may show the connection between metabolites utilization and cell motility.

**(ii) Differential expression analysis.** DESeq analysis identified differentially upregulated and downregulated genes under specific conditions. Pathways involved in general metabolism such as "biosynthesis of amino acids," "ABC transporters," and "two-component system" were significantly enriched in at least 24 out of 25 pairwise comparisons (Fig. S11a). The genes in these pathways were likely to play a more prominent role in stress adaptation. In contrast, some pathways demonstrated differential expression in very few pairwise comparisons. These pathways included "beta-lactam resistance" and "tyrosine metabolism" that were enriched in only two and one pairwise comparison, respectively (Fig. S11b). Hence, these pathways may be less affected by stress conditions.

**(iii) Genome-wide target prediction.** IntaRNA calculated binding energies between each sRNA and all other genes (predicted sRNAs and annotated genes). The binding energy distribution enabled the calculation of the *P*-values and FDR (false discovery rate) for each sRNA-target interaction. Among interactions with FDR 0.05, CjSA21 was the only sRNA with targets showing KEGG pathways enrichment. The enriched pathways were "bacterial chemotaxis" (FDR = 5.65E-10) and "two-component system" (FDR = 1.16E-08).

**(iv) Combined results.** After combining the results from WGCNA, DESeq2, and IntaRNA, 513 sRNA-target interactions (Fig. S12) fulfilled all three criteria for our RNA-RNA interactions (described in Fig. S9a legend). To identify sRNAs that may act as critical regulators of particular biological pathways, KEGG pathways enrichment analysis on the targets of each sRNA enabled the identification of potential regulatory roles for each sRNA. sRNA with targets over represented in at least one pathway are shown in Table 1. In particular, all four CjSA21 targets were found to be involved in bacterial chemotaxis and the two-component system and showed the lowest FDR values. CjSA28, CjSA53, and CjSA9 also had targets that showed KEGG pathway enrichment. However, not all of their targets belong to the enriched pathways.

**Detailed analysis of CjSA21.** To illustrate the potential of deriving sRNA-mRNA interactions using our RNAtag-seq data, CjSA21 was selected for further evaluation as CjSA21 targets showed the most statistical KEGG pathways enrichment for both the integrative analysis and IntaRNA. Hence, CjSA21 was an outstanding example for a more detailed analysis below.

CjSA21 is situated downstream of *rpmI* and *rplT*, antisense to Cj0246c, and is potentially under the regulation of the TSS antisense to Cj0246c. Furthermore, expression coverage (Fig. S13) suggested that CjSA21 also formed an operon with *rpmI* and *rplT*. Hence, CjSA21 might be usually regulated together with *rpmI* and *rplT* using the primary TSS of *rpmI*. Evidence suggested that CjSA21 might also express independently from the alternative TSS under certain conditions like hyperosmotic shock (nacl). (Fig. S9c, 13).

**TABLE 1** All sRNAs with at least three targets being found in a KEGG pathway, with FDR (for KEGG pathways enrichment) ≤ 0.05[a]

| Sample | Targets | KEGG pathways | Hits | FDR |
|---|---|---|---|---|
| CjSA21 | 4 | Bacterial chemotaxis | 4 | 5.08E-08 |
| CjSA21 | 4 | Two-component system | 4 | 4.71E-07 |
| CjSA28 | 14 | Bacterial secretion system | 3 | 2.25E-04 |
| CjSA53 | 19 | Purine metabolism | 4 | 3.58E-03 |
| CjSA53 | 19 | Pyrimidine metabolism | 3 | 1.09E-02 |
| CjSA9 | 10 | Purine metabolism | 3 | 3.76E-03 |

[a]Only KEGG pathways with less than 100 proteins were shown here.

Further investigation of CjSA21 targets illustrated their importance in chemosensing. As shown in Fig. S9b, the four targets of CjSA21 are Cj1506c (*tlp1*), Cj0144 (*tlp2*), Cj1564 (*tlp3*), and Cj0262c (*tlp4*). Because these four genes encoded Tlps, CjSA21 may regulate chemotaxis and signal transduction (Table 2).

**(i) Negative coexpression between CjSA21 and *tlp1* to *4*.** CjSA21 belongs to module II, which is comprised of chemotaxis and metabolites utilization genes. The coexpression of module II is negatively correlated with food processing conditions (nacl, 5% ce and cold) and positively correlated with starvation and the early stationary phase at 42°C (42 ES), iron-limitation (iron lim ES), and iron-repletion (iron rep ES) (Fig. 3a and 4a). Both "two-component system" and "bacterial chemotaxis" were also among the differentially enriched pathways that appeared in 24 and 17 out of 25 pairwise comparisons, respectively, suggesting their importance in *C. jejuni* stress adaptation.

Enrichment analysis of module II genes showed that the most significantly enriched KEGG pathways were "bacterial chemotaxis" and "two-component system," as module II includes genes such as *tlp1* to *4*, *cheA*, and *cheV*. Notably, module II was also enriched with amino acids and fatty acids biosynthetic pathways (Fig. 4b; Fig. S14). A comparison of variance stabilizing transformation (vst) expression between CjSA21 and *tlp1* to *4* demonstrated statistically significant bicor coefficients between −0.567 and −0.794 (Fig. 4c).

We subdivided each coexpression module into two clusters with opposing vst expression values and differential expression patterns (Fig. S15, 16). In module II, cluster 1 consisted of *tlp1* to *4* and other genes from several other pathways, including

**TABLE 2** All experimental conditions for Cappable-seq and RNAtag-seq[a]

| Sample name | Initial growth | Treatment[b] |
|---|---|---|
| 37 M | Exponential phase at 37°c | NA |
| 37 ES | Early stationary phase at 37°c | NA |
| 37 LS | Late stationary phase at 37°c | NA |
| 42 M | Exponential phase at 42°c | NA |
| 42 ES | Early stationary phase at 42°c | NA |
| 42 LS | Late stationary phase at 42°c | NA |
| Cold | Exponential phase at 37°c | Resuspended in MH2 broth at 4°C for 24 h (78) |
| 5% ce | Exponential phase at 37°c | Incubated in MH2 broth supplemented with 5% chicken exudate at 4°C for 24 h (79) |
| Acid | Exponential phase at 37°c | Resuspended in MH2 broth at pH 3.5 for 10 min (80) |
| Ana | Exponential phase at 37°c | Incubated in anaerobic chamber for 1 h |
| Heat | Exponential phase at 37°c | Incubated at 55°C for 3 min (81) |
| Iron lim M | Exponential phase at 37°c | Growth media was MEM$\alpha$ supplemented with 10 M pyruvate (37) |
| Iron lim ES | Early stationary phase at 37°c | Growth media was MEM$\alpha$ supplemented with 10 M pyruvate (37) |
| Iron rep M | Exponential phase at 37°c | Growth media was MEM$\alpha$ supplemented with 10 M pyruvate and 40 M FeSO$_4$ (37) |
| Iron rep ES | Early stationary phase at 37°c | Growth media was MEM$\alpha$ supplemented with 10 M pyruvate and 40 M FeSO$_4$ (37) |
| Nacl | Exponential phase at 37°c | Incubated in 1.5% NaCl for 2 h (82) |
| Oxidative | Exponential phase at 37°c | Added 3 mM H$_2$O$_2$ for 10 min (83) |
| Starv | Early stationary phase at 37°c | Resuspend in Ringer's solution for 5 h (84) |
| GSNO | Exponential phase at 37°c | Incubate in 1.5 mM GSNO for 2 h (85) |
| Sod deoxy M | Exponential phase at 37°c | Growth media was supplemented with 0.1 % sodium deoxycholate (86) |
| Sod deoxy ES | Early stationary phase at 37°c | Growth media was supplemented with 0.1 % sodium deoxycholate (86) |

[a]The cells were cultured in MH2 broth unless specified otherwise.
[b]NA, standard growth conditions; no treatment.

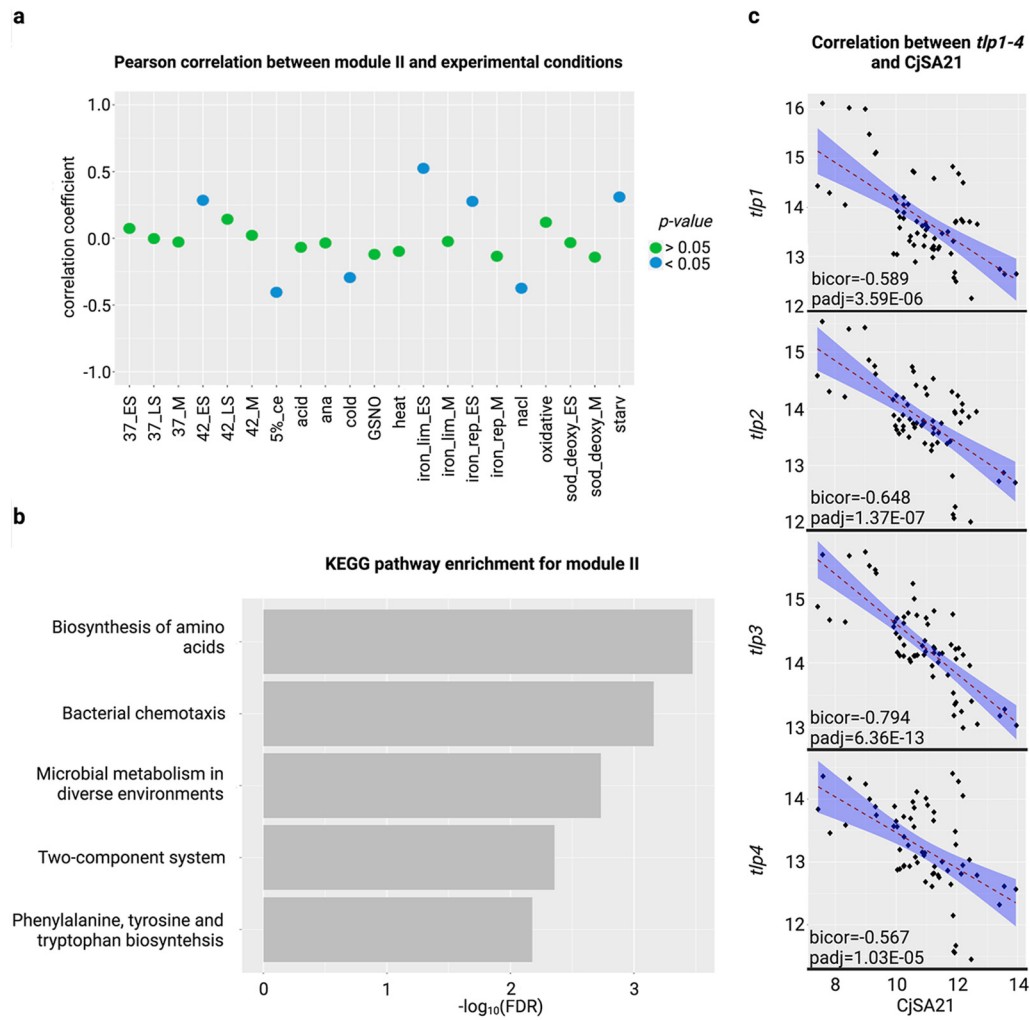

**FIG 4** (a) The Pearson correlation coefficient and *P*-values between the eigengene (the first principal component of the gene expression matrix) of modules II and experimental traits. (b) KEGG pathways enrichment on module II. Only the five pathways with the lowest FDR were shown here. (c) The correlation between CjSA21 and *tlp1* to *4* across all 63 RNA-Seq replicates.

"propanoate metabolism," "lysine biosynthesis," and "cysteine and methionine metabolism." High vst expression values and differential upregulation were observed under early stationary and starvation conditions.

Cluster 2 consisted of CjSA21 and genes involved in the following pathways "phenylalanine, tyrosine, and tryptophan biosynthesis" and "glycine, serine, and threonine metabolism." Both pathways consist of members of the tryptophan biosynthesis (Trp) operon. CjSA21 and the Trp operon showed high expression values and differential upregulation under food processing conditions (cold, 5% ce and nacl). Trp operon was also differentially upregulated in other stress conditions, including heat stress (heat), anaerobic stress (ana), and nitrosative stress (GSNO).

**(ii) Opposite differential expression patterns under food processing conditions.** While coexpression analysis showed the overall negative correlation between CjSA21 and *tlp1* to *4*, understanding the stress adaptation role of CjSA21 required understanding its differential expression under conditions of interest (Table S3). Fig. 5a displayed the differential expression of CjSA21 and the four *tlp* genes across all selected pairwise comparisons. The results showed the most apparent opposite expression pattern under food processing conditions (cold, 5% ce and nacl) with standard laboratory control (37 M) as the control. All four *tlp* genes demonstrated differential downregulation and vice versa

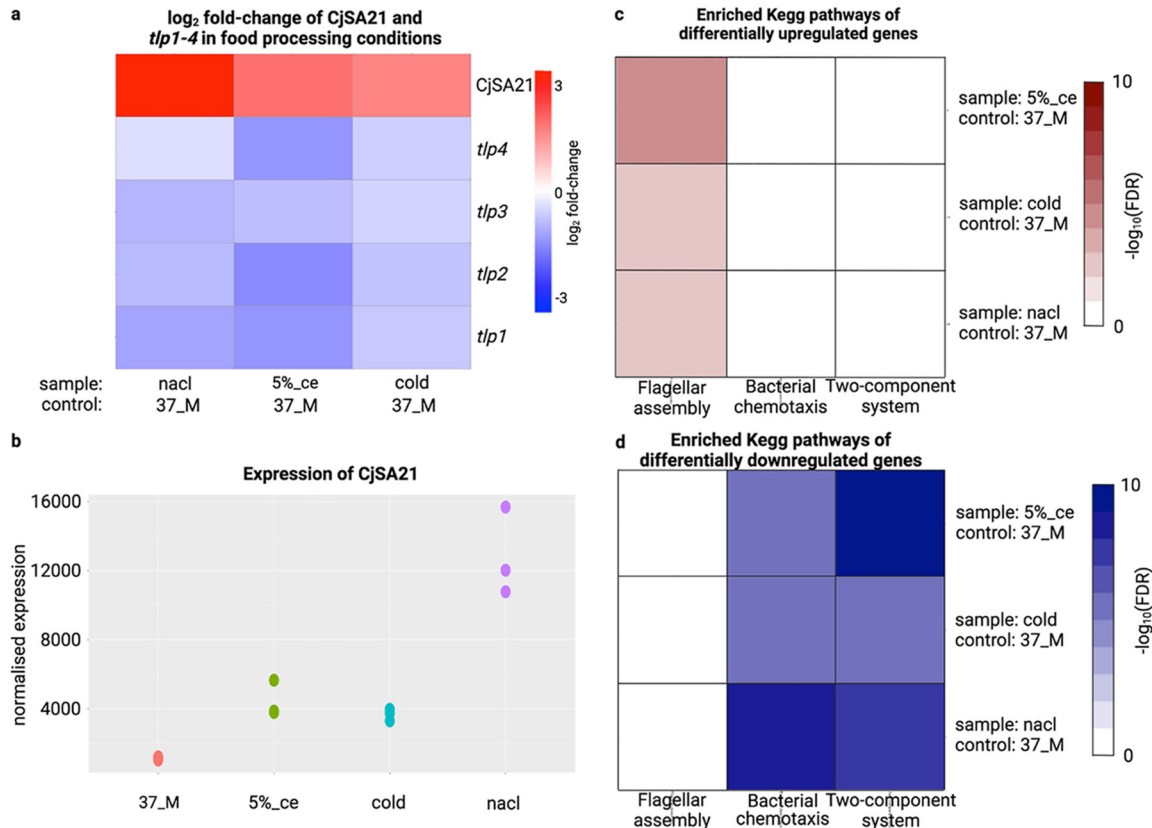

**FIG 5** (a) log$_2$-fold change of CjSA21 and *tlp1* to *4*. Red color. (b) normalized expression values of CjSA21 under 37 M, cold, nacl, and 5% ce. The expression values were normalized using DESeq's median to ratio (c) KEGG pathway enrichment of differentially upregulated genes (d) KEGG pathway enrichment of differentially downregulated genes. Only "bacterial chemotaxis, "flagellar assembly," and "two-component system" were shown here. Values with FDR > 0.05 were colored in white.

for CjSA21. In other pairwise comparisons, not all four *tlp* genes showed opposite differential expression patterns (Fig. S17). CjSA21 demonstrated the highest expression level under nacl and a modest increase under cold and 5% ce (Fig. 5b; Fig. S18).

Under the three pairwise comparisons involving food processing conditions, KEGG pathway enrichment analysis showed statistical overrepresentation of "bacterial chemotaxis" and "two-component system" among differentially downregulated genes (Fig. 5c). Interestingly, differentially upregulated genes exhibited significant FDR for "flagellar assembly" in all three comparisons (Fig. 5c, d).

**(iii) Genome-wide target predictions.** Negative coexpression and opposite differential expression patterns may be due to coregulation rather than direct sRNA-mRNA binding. We explored the likelihood of direct RNA-RNA interactions using genome-wide target prediction by IntaRNA. Among all CjSA21 interactions with *P*-values 0.05, most energy values ranged between −20 to −40 kcal/mol. Interestingly, all four of *tlp1* to *4* exhibited binding energy of around −45 kcal/mol, with significant *P*-values and FDR (Table 3; Fig. S19a). *tlp9* and *tlp10* can also form strong interactions (FDR 0.05) with

**TABLE 3** All CjSA21 targets predicted by IntaRNA$^a$

| Gene ID | Gene name | Binding energies (kcal/mol) | *P*-value | FDR |
|---|---|---|---|---|
| Cj0144 | *tlp2* | −44.91 | 1.97E−05 | 5.71E−03 |
| Cj0262c | *tlp4* | −44.91 | 1.97E−05 | 5.71E−03 |
| Cj1506c | *tlp1* | −45.69 | 1.52E−05 | 5.71E−03 |
| Cj1564 | *tlp3* | −44.91 | 1.97E−05 | 5.71E−03 |
| *cetA* | *tlp9* | −47.73 | 7.70E−06 | 5.71E−03 |
| Cj0019c | *tlp10* | −43.2 | 3.47E−05 | 8.62E−03 |

$^a$Only interactions with FDR ≤ 0.05 are shown here.

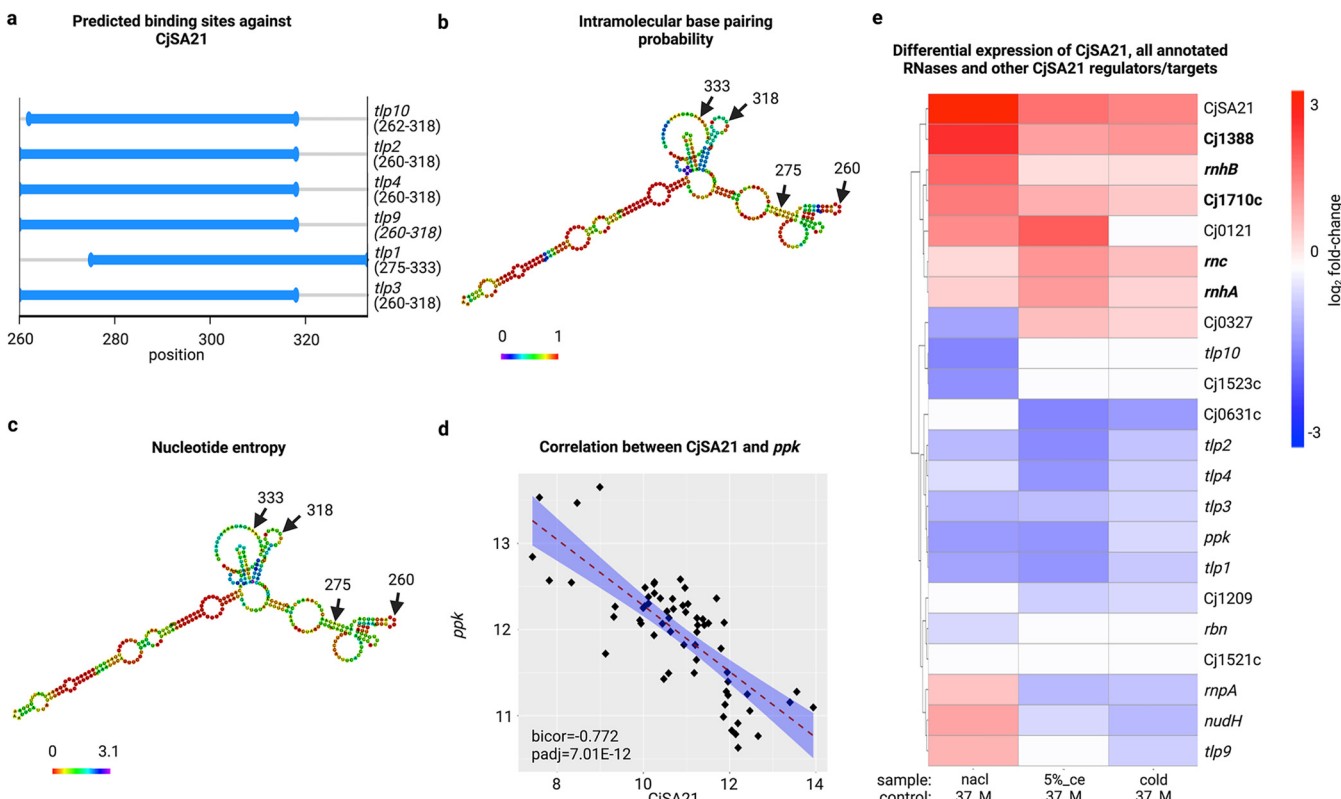

**FIG 6** (a) The distribution of optimal binding sites of CjSA21 against *tlp1* to *4*, *tlp9*, and *tlp10*. (b, c) The intramolecular binding probability and entropy of CjSA21 predicted by RNAfold. (d) Pairwise correlation between the vst values of *ppk* and CjSA21. (e) log₂-fold change of CjSA21, *ppk*, *tlps*, and all annotated RNases. This figure only showed the pairwise comparisons with 5% ce, nacl, and cold compared against 37 M. Differentially expressed RNases under all three conditions were highlighted in bold text.

CjSA21. All CjSA21-*tlp* interactions had binding energy values at around −45 kcal/mol, which was more stable than other putative CjSA21-mRNA interactions with binding energy > −20 kcal/mol (Fig. S19b).

All six genes targeted the regions near the 3′-end of CjSA21. *tlp2*, *tlp3*, and *tlp4* shared identical binding sequences, while *tlp10* was only different by two bp (Fig. 6a). However, neither *tlp10* nor *tlp9* belonged to the same coexpression module as *tlp1* to *4* (module II). Compared with *tlp1* to *4*, *tlp9*, and *tlp10* had a weaker correlation with CjSA21 expression (Fig. S20). Also, both *tlp9* and *tlp10* displayed opposite differential expression with CjSA21 in only one of three food processing conditions (cold, 5% ce and nacl with 37 M as controls) (Fig. S21). CjSA21 may regulate the translation or stability of *tlp1* to *4* without affecting its transcript abundance.

**(iv) Structural and expression analysis identified proteins that might mediate CjSA21-*tlp* interactions.** Structural prediction by RNA-fold suggested that the binding sequence of CjSA21 was located in regions that contained both paired and unpaired bases. The region also exhibited high nucleotide entropy and low intramolecular binding probability (Fig. 6b, c). The results suggested that CjSA21 targeted *tlp1* to *4* using regions with less stable secondary structures.

Published work suggested the gene encoding Polyphosphate kinase (*ppk*) mediated *C. jejuni* osmotic stress survival and biofilm formation (48). Here, *ppk* exhibited strong negative coexpression against CjSA21 (bicor = −0.792) and an opposite differential expression pattern under several conditions, including hyperos-motic stress, cold stress, and 5% chicken exudate (Fig. 6d). However, the IntaRNA-calculated binding energy between *ppk* and CjSA21 was statistically insignificant. This observation suggested *ppk* might be coregulated with CjSA21. It was also possible that *ppk* directly or indirectly inhibited transcription of CjSA21.

Because the predicted binding site was near the mRNA 3′-ends, we speculated that CjSA21 inhibited *tlp1* to *4* expression by RNase degradation instead of RBS blockage. Another question was which RNases were more likely to mediate the degradation of *tlp1* to *4*. Fig. 6e showed the differential regulation of all annotated RNases, alongside CjSA21, *ppk*, *tlp1* to *4*, and *tlp9* to *10* for comparison. It was noteworthy that the following RNases were also found in module II: Cj0121, *rnc*, *rnhA*, Cj1710c. All except Cj0121 displayed differential upregulation in the three conditions of interest (highlighted in bold text in Fig. 6e). All four RNases in module II showed moderate (below −0.3) to strong negative coexpression (below −0.85) with *tlp1* to *4* (Fig. S22). Also differentially upregulated in the three food processing conditions were RNases Cj1388 and *rnhB*. However, Cj1388 was not inversely coexpressed with *tlp1* to *4*. *rnhB* showed weak/moderate negative coexpression against *tlp1* to *4* (−0.35 to −0.527) (Fig. S23). All these indicated that Cj0121, *rnc*, *rnhA*, Cj1710c, and *rnhB* were the most likely candidates that degraded *tlp1* to *4* transcripts without digesting CjSA21.

## DISCUSSION

This study presents one of the most comprehensive transcriptomic data sets in *C. jejuni*, which allowed the discovery of 3,613 novel TSS and condition-specific gene expression across 21 conditions. This data set represents a valuable resource for the wider *C. jejuni* research community to investigate the *C. jejuni* transcriptomic landscape changes. Our data set covers 21 conditions and enables simultaneous comparisons of these stress conditions. The generated transcriptomic data were used to detect novel sRNAs and their potential binding targets and construct the global sRNA-mRNA interactome unprecedented for *C. jejuni*. As an example, CjSA21 was established as a novel inhibitor of Tlp chemosensing under food processing conditions. The transcriptional profile also revealed upregulation of flagellar biosynthesis and downregulation of *ppk*. All of these hint toward increased biofilm formation (49–52), which promotes survival under food processing conditions. Further gene expression analysis has suggested several RNases and *ppk* are also involved in the CjSA21 regulatory mechanism. This study has illustrated how our data set can uncover sRNA sequence and biological activities. It has used CjSA21 as an example that might improve food safety. This study has also predicted 513 sRNA-target interactions as candidates for experimental validation.

In this study, 3,613 novel TSS were identified using Cappable-seq. This outcome indicates that Cappable-seq is more sensitive to TSS detection compared with dRNA-seq, possibly because Cappable-seq does not rely on 5′-monophosphate dependent terminator RNA exonuclease (TEX), which is prone to inhibition by secondary structures (41). The difference of TSS positions among published results might result from different experimental methodologies (35, 36, 39). For instance, cells were harvested in different $OD_{600}$ values among the three studies. Moreover, Porcelli et al.'s (36) samples were sequenced by the Roche 454 platform, while the data of Dugar et al. (35) and Handley et al., 2015 were generated from the Illumina HiSeq platform. Also, RNA samples from nonstandard conditions revealed condition-specific TSS. MEME motif analysis has highlighted the additional upstream guanosine for sigma-70, which contributes to transcription activity of weak sigma-70 promoters in *Escherichia coli* (53). Another interesting finding is the abundance of novel internal TSS. While some might result from spurious transcription from the compact AT-rich genome, others may contribute to condition-specific transcripts derived from a coding region. Antisense TSS also occurs frequently among sigma-70 and sigma-28 promoters. Interestingly, most of the sigma-28 antisense TSS were on the opposite strand of metabolic pathways other than flagella-associated genes. That may suggest that sigma-28 regulates metabolic pathways other than flagellar assembly.

This study has produced the first RNA-Seq data set obtained from 21 experimental conditions. Such a comprehensive data set is valuable for understanding *C. jejuni* growth and virulence. For instance, *C. jejuni* is avirulent in chickens but pathogenic in humans and conditions that model and reflect the chicken and human host. Samples grown under 37°C and 42°C helped investigating the host-dependent variation of

virulence. In addition, data from the cold stress 4°C can help understand how *C. jejuni* can survive in refrigerated poultry meat. Other conditions can elucidate the adaptation strategy along the gastrointestinal tract, where the cells encounter varied oxygen content, metal iron availability, pH, and bile salt concentration. While individual studies have looked into these conditions, those studies were conducted separately with different culturing conditions and experimental setups. In contrast, this data set provides valuable information for simultaneous comparison of all these conditions with the same culturing conditions and consistent experiment setups.

The ANNOgesic results showed that more than one in every four sRNAs predictions (including CjSA21) came from data from this study. Like the previous conservation analysis (35), most sRNAs only showed sequence conservation among *C. jejuni* strains. Such a pattern suggests that the *C. jejuni* genomes may carry a unique repertoire of sRNAs. In addition, over two thirds of the sRNAs have not been identified from previous publications. Hence, further investigation of *C. jejuni* sRNAs is likely to expand our understanding of other bacterial sRNA sequences, structures, and functions.

In this study, 513 sRNA-target interactions were extracted by combining WGCNA, DESeq, and IntaRNA. Key sRNA-target interactions were selected by similarly searching for KEGG pathways enrichment as rNAV 2.0 (54). Notably, all four CjSA21 targets are Tlps that transduce external stimuli that lead to a chemotactic response. Interestingly, CjSA21 is highly conserved among both *Helicobacter* and *Campylobacter* strains, suggesting that CjSA21 might be a conserved chemotaxis regulator among Epsilonproteobacteria. Interestingly, *Helicobacter pylori* RepG sRNA also binds to *tlpB* mRNA and regulates TlpB translation (43, 55). Hence, chemotaxis regulation by sRNA might be conserved among Epsilonproteobacteria.

**Conclusion.** In summary, we have generated transcriptomic data in *C. jejuni*, which allows simultaneous comparison of transcription landscapes across both standard and nonstandard laboratory conditions. This transcriptional landscape will enable the wider *C. jejuni* research community to understand stress adaption by sRNA expression better. A detailed analysis has further identified CjSA21 as a potential inhibitor of chemosensing for surviving under food processing conditions. Further exploration of this data set will aid in the discovery of a broader range of sRNA regulatory mechanisms and cross talk between pathways.

## MATERIALS and METHODS

**Maintenance and growth of *C. jejuni*.** The strain used in this study is the original clinical motile isolate of NCTC 11168 from the National Collection of Type Cultures. The strain was routinely cultured on *Campylobacter* blood-free selective agar supplemented with CCDA (containing Cefoperazone and Amphotericin B) (Oxoid) under microaerophilic conditions in a variable atmosphere incubator (VAIN) (Whitley VA500 workstation cabinet) (90% N2 [vol/vol], 6% CO2 [vol/vol], 4% O2 [vol/vol]) at 37°C. After 24 h, NCTC 11168 was subcultured onto agar plates containing cation-adjusted Mueller-Hinton 2 (MH2) (Sigma-Aldrich) 1.5% agar (Bacto Agar) and incubated overnight. From MH2 plates, a loop of NCTC 11168 culture was resuspended in 5 mL of MH2 broth in 25 cm³ Vented Capped Tissue culture flasks (Falcon) shaking at 200 rpm on an orbital shaker (Ika, VXR basic Vibrax) in the VAIN for 15 to 15 h, 30 min.

The optical density at 600 nm ($OD_{600}$) was measured on a spectrophotometer (Biochrom), and the calculated volume of liquid culture to achieve an initial $OD_{600}$ of 0.05 was centrifuged at 5,000 $\times$ *g* for 3 min and resuspended in 4.5 mL of fresh MH2 broth in a 6-well plate (Greiner).

Table 2 summarizes all experimental conditions used in this study. More details on all conditions can be found in the Supplementary Methods file.

The details of RNA extraction, RNA processing and sequencing library preparation are in the Supplementary Methods file.

**Cappable-seq data analysis.** The Cappable-seq data were analyzed following the scripts and instructions in https://github.com/Ettwiller/TSS (Fig. 1a). Fifty nucleotides upstream of determined TSS were extracted to search for consensus promoter motifs in MEME version 5.0.5 (https://meme-suite.org/meme/) (56). Parameters included selecting the "search given strand only" option and setting the maximum motif size as 50 nucleotides. The exact numbers of each motif were determined using the matchPattern() function of the R package BioStrings.

**RNAtag-seq data analysis.** Raw data of RNAtag-seq samples were demultiplexed using an in-house script (see GitHub page for more details). The first 9 bp of Read 1 containing the unique tag/barcode was trimmed manually and aligned to the NCTC11168 reference genome (Accession: NC 002163.1) before undergoing RNA-seq analysis as described on the GitHub page.

**sRNA prediction.** sRNAs were detected from the *C. jejuni* NCTC11168 genome sequence (Accession: NC 002163.1). The following tools were used: RNAdetect (57), RNAz 2.0 (58), RNAz (59), a SVM tool based on trinucleotide composition (60), and StructRNAfinder (61). The consensus made by at least three of the tools (designated "Genome prediction") was carried forward for further evaluation.

Published transcriptomic data sets (35–37, 39, 62–64) and data from this study were used for sRNA prediction.

ANNOgesic detected sRNAs using wiggle files created from bam2wig.py (Galaxy Version 2.6.4) (65). All BAM files were normalized with specified wigsum = 100000000. The transcript 5′-ends were defined using both the mentioned wig files and TSS positions from Cappable-Seq and published TSS data sets (35, 36, 39). Putative transcripts with normalized coverage values below 80 were filtered out to minimize false positives. The toRNAdo source code and instructions are available on https://github.com/pavsaz/toRNAdo.

The conservation of putative sRNAs was analyzed by searching homologous sequences using the NCBI BLASTN suite. Sequences of query sRNAs were compared with a database comprised of genomes from Epsilon-proteobacteria strains in the conservation analysis in Dugar et al. (35). Putative sRNAs and transcriptomic coverage were visualized using Integrative Genomics Viewer (IGV) version 2.8.13 (66).

**Integrative bioinformatics analysis. (i) Coexpression analysis.** Gene coexpression analysis was carried out using Weighted Correlation Network Analysis (WGCNA) R packages version 1.69 (67) with the following nondefaulted parameters: (i) The correlation matrix was constructed using the biweight mid-correlation (bicor) to minimize the disruption from outliers. (ii) Raw read counts were normalized by the vst function of the DESeq2 package (68) to ensure similar read distribution across samples. (iii) Based on the scale-fit topology fit and mean connectivity, the bicor correlation matrix was power transformed with soft-threshold power = 5 to construct an adjacency network. (iv) An unsigned adjacency network was constructed instead of a signed adjacency network to allow equal treatment between negative and positive correlations (Fig. S1).

**(ii) Differential expression analysis.** The mapped reads were annotated using CoverageBed from Bedtools v2.27.1 (69). The differential gene analysis was conducted with the DESeq2 package version 1.22.2 (68). Default settings were used for both Bedtools and DESeq2.

**(iii) KEGG pathway functional enrichment analysis.** KEGG Pathway enrichment of each coexpression module and pairwise differential expression was analyzed using the Bioconductor package STRINGdb version 1.22.0, using the default settings (70, 71).

**(iv) Genome-wide sRNA target prediction.** The binding affinity between each sRNA-target duplex was calculated using IntaRNA version 3.2.0, using the optimized parameters for sRNA-target predictions (72).

**(v) Network visualization.** RNA-RNA networks were visualized in Cytoscape version 3.8.2 (73), with Cytoscape images generated automatically using the Bioconductor package version 2.2.9 (74).

**(vi) Structural prediction and analysis.** RNA structural prediction was performed in the RNAfold web server (75–77) using the default parameters.

**Source code.** All scripts for this study are available https://github.com/StephenLi55/c.-jejuni-integrative-analysis unless specified otherwise.

**Data availability.** Transcriptomic data can be found in the ARRAYEXPRESS database (www.ebi.ac.uk/arrayexpress). The accession number for Cappable-seq and RNAtag-seq are E-MTAB-11308 and E-MTAB-11309, respectively.

## SUPPLEMENTAL MATERIAL

Supplemental material is available online only.
**SUPPLEMENTAL FILE 1**, PDF file, 1.2 MB.

## ACKNOWLEDGMENTS

The authors thank Emma Denham, Sascha Ott, Meera Unnikrishnan, Alexia Hapeshi, Emily Stoakes, and Nigel Dyer for their advice and discussion that have contributed to the work in this study. S.L. was supported by the Medical Research Council (MRC) Doctoral Training Program (DTP) from the National Productivity Investment Fund (NPIF) (Grant code: MR/R502212/1) and the Medical and Life Sciences Research Fund (MLSR). J.L. was supported by the Biotechnology and Biological Sciences Research Council (BBSRC) Midlands Integrative Biosciences Training Partnership (MIBTP) (Grant code: BB/M01116X/1). M.T.A. is supported by the United Arab Emirates University start-up grant (G00003688).

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
