## [Reviewer comments · Microbiology Spectrum]

Microbiology Spectrum

Post-transcriptional regulation in response to different environmental stresses in *Campylobacter jejuni*

Stephen Li, Jenna Lam, Leonidas Souliotis, Mohammad Alam, and Chrystala Constantinidou

Corresponding Author(s): Chrystala Constantinidou, University of Warwick

Review Timeline:

Submission Date:	January 25, 2022
Editorial Decision:	March 4, 2022
Revision Received:	May 4, 2022
Accepted:	May 10, 2022

Editor: Amanda Oglesby

Reviewer(s): Disclosure of reviewer identity is with reference to reviewer comments included in decision letter(s). The following individuals involved in review of your submission have agreed to reveal their identity: Victor J. DiRita (Reviewer #2)

Transaction Report:

DOI: <https://doi.org/10.1128/spectrum.00203-22>

March 4, 2022

Dr. Chrystala Constantinidou
University of Warwick
Microbiology and Infection Unit
Warwick
United Kingdom

Re: Spectrum00203-22 (Post-transcriptional regulation in response to different environmental stresses in *Campylobacter jejuni*)

Dear Dr. Chrystala Constantinidou:

Thank you for submitting your manuscript to Microbiology Spectrum. Your manuscript was evaluated by two experts, and based on their critiques I have decided that your manuscript is not suitable for publication in its current form. Both reviewers commented on the utility of the datasets for the field, but they also noted concerns regarding the recognition of previous work, presentation of methodologies, and interpretation of results. I would be happy to consider a revised manuscript that addresses these concerns.

Link Not Available

Sincerely,

Amanda Oglesby

Journals Department
Reviewer comments:

Reviewer #1 (Comments for the Author):

This manuscript contains one of the most comprehensive analyses of the environmental regulation of gene expression in *Campylobacter jejuni*. One of the major findings is of a large number of previously unrecognised sRNA and data indicating that these sRNAs regulate important phenotypes in an environment specific manner. This data is likely to be highly influential in managing human infections as this organism is the major agent of foodborne gastroenteritis in developed countries and the

examined conditions are relevant to the sources of infection including spread through the foodchain.

A key feature of this article is the detection of sRNA by combinations of sequence-based analyses and detection of transcripts/transcriptional start sites. These approaches appear to have been successful in detecting a number of novel sRNAs. There is also a good prediction of sRNA operons through target prediction software with interesting downstream validation and exploration of co-expression under different environmental conditions. In particular, analysis of CjSA21 is detailed and has clear indications of a regulon consisting of chemotaxis genes, negative regulation by *ppk* and a mechanism involving RNA degradation of the target transcripts. Another feature is the highly sensitive detection of TSS in comparison to previous studies presumably as a result of the different methodology utilised.

The re-analysis of the sigma factors produces some modifications of the potential bindings sites. It was not clear from the presented data however whether the consensus sequence for the sigma factors is different between the primary/secondary motifs and other locations. It would also be useful to know if there is a wide variation in expression levels between different sites and if this correlates with motif.

Parts of this manuscript were however difficult to understand with many of the results section lacking any introduction and assumptions of highly detailed knowledge of the methodologies. The most important of these aspects are listed under major changes and should be addressed prior to publication if this article is going to be widely accessible.

Major changes

The start of the results section describes results of TSS in previous studies. There is however no indication of species (presumably *C. jejuni*) or strain (all NCTC11168?) studied in these publications or what the environmental stresses were where this was examined. There also huge differences between the studies that are unexplained but presumably relate to methodology.

Lines 196-199. These sentences are unclear with loose descriptions of the associations (nearby) and numbers (most, Some, several). These sentences I assume relate to the 116 sRNAs predicted by ANNOgesic but this is not entirely clear. Predictions of 5' ends should have some indications of distances so it is clear that they are reasonable.

Lines 196-211. These paragraphs discuss the association of TSS with sRNA. Line 211 indicates that the TSS data was from all conditions which indicates that some sRNAs may have been missed by previous studies due to only using standard conditions. However the descriptions of the data do not give any indications of how many of the TSS were found under non-standard conditions and hence it is not possible to properly assess if this is the reason that previous studies did not find them.

Lines 227-228 refer to 11 expression modules. The data used for derivation of these modules is not clear. Were these predictions made using the expression data from all of the different conditions or only some? If so wouldn't the modules vary depending on which conditions are used? Further, it is not possible to evaluate the nature of these modules as no data is provided on the numbers of genes in each module or indeed a list of the specific genes. Some indications of module content are provided but this is gleaned as the text progresses and is somewhat frustrating and hinders understanding. More detailed data on the modules will be required if other groups are going to generate comparative data or indeed to do further confirmatory analyses.

Minor

Line 38-39, colloquial use of lab and unclear use of transmission chain

Table 1. Could be more informative. Treatment could be split into two columns with initial growth conditions indicated in one column and incubations conditions in the second. Periods of incubation in specific conditions need to be indicated e.g. for 'acid' how long were cells subjected to incubation in MH2 broth at pH3.5.

Figure 3. Legend: misspelling 'a person correlation'

Reviewer #2 (Comments for the Author):

Li et al. present a comprehensive dataset of RNASeq analysis of *Campylobacter jejuni* (strain 11168) grown under 21 different in vitro conditions. The conditions selected are defined by the authors as "host and transmission relevant", and include those emphasizing temperature, acid, iron limitation, starvation, and many others. The major thrust of the work is aimed in particular at identifying transcription start sites and small non-coding RNAs that might contribute to fitness in these conditions. After some quality-control and filtering criteria were applied, 96 sRNAs were ultimately analyzed further. The sRNAs were categorized into 11 modules based on patterns of co-expression, which were further classified to some degree by KEGG pathway associations within co-expression modules. One sRNA, CjSA21 was selected for further analysis, and putative targets for this were identified using the RNA:RNA interaction tool IntaRNA. This led the authors to suggest that CjSA21 might target the RNA for the chemotaxis motility transducer protein Tlp1-4, and co-expression of these RNAs occurred under 'food-processing' conditions (e.g., cold temperatures, presence of 'chicken exudate'), leading the authors to further speculate that this interaction might

promote elevated biofilm growth (a sessile as opposed to motile lifestyle) during food processing.

This study represents a significant amount of work and valuable analysis of RNASeq data in an important foodborne pathogen. It provides a roadmap for others who might seek to do similar comprehensive analysis of sRNA function, and the experimental and bioinformatic approaches used by these authors will certainly be instructive to others. The major weakness comes from the highly speculative conclusions drawn by the authors based - for the most part - on co-expression and InstRNA findings, with very little experimental follow-up to test any specific hypotheses. While a weakness, this is not a major flaw in the work, but rather a statement of its limitations.

Specific comments follow

1. To be more rigorous in their scholarship, the authors must clarify their description of previous transcriptomic work on *C. jejuni*. They state in their abstract that "published *C. jejuni* sRNAs were discovered during growth under standard laboratory conditions..." Earlier work they cite (Taveirne et al) reported significant findings regarding sRNA expression from a transcriptomic analysis of *C. jejuni* 81176 harvested from experimentally-infected chickens. This needs to be clearly described in their rationale for this study, as the way they describe prior work in this area is misleadingly narrow.

Related to this point, the authors should compare the sRNAs they identified in this study to those described in the experiments from the chicken infection - to what degree do the two datasets (in vitro v. in vivo) overlap? The major comparisons carried out in this study are to RNASeq from other in vitro conditions.

2. A significant amount of the data in this paper is reported in supplemental files and that should be reconsidered. Figures S8 and S11, for examples, should be included along with the main figures and not in supplemental material.

Staff Comments:

Preparing Revision Guidelines

Please return the manuscript within 60 days; if you cannot complete the modification within this time period, please contact me. If you do not wish to modify the manuscript and prefer to submit it to another journal, please notify me of your decision immediately so that the manuscript may be formally withdrawn from consideration by Microbiology Spectrum.

Reviewer #1 (Comments for the Author):

This manuscript contains one of the most comprehensive analyses of the environmental regulation of gene expression in *Campylobacter jejuni*. One of the major findings is of a large number of previously unrecognised sRNA and data indicating that these sRNAs regulate important phenotypes in an environment specific manner. This data is likely to be highly influential in managing human infections as this organism is the major agent of foodborne gastroenteritis in developed countries and the examined conditions are relevant to the sources of infection including spread through the foodchain.

A key feature of this article is the detection of sRNA by combinations of sequence-based analyses and detection of transcripts/transcriptional start sites. These approaches appear to have been successful in detecting a number of novel sRNAs. There is also a good prediction of sRNA operons through target prediction software with interesting downstream validation and exploration of co-expression under different environmental conditions. In particular, analysis of CjSA21 is detailed and has clear indications of a regulon consisting of chemotaxis genes, negative regulation by ppk and a mechanism involving RNA degradation of the target transcripts. Another feature is the highly sensitive detection of TSS in comparison to previous studies presumably as a result of the different methodology utilised.

We would like to thank the reviewer for their positive assessment of our work. The reviewer has raised some very important questions which have been addressed below.

The re-analysis of the sigma factors produces some modifications of the potential bindings sites. It was not clear from the presented data however whether the consensus sequence for the sigma factors is different between the primary/secondary motifs and other locations. It would also be useful to know if there is a wide variation in expression levels between different sites and if this correlates with motif.

We apologise for the unclear description of the data. The consensus sequence motif of each sigma factor was used to search for motifs upstream of all TSS regardless of TSS type. The distribution of sigma motifs among different TSS categories is visually presented in Figure 1d and we have included additional information in the main text to address your points.

“The frequency of sigma-70 and sigma-54 motifs among Cappable-seq TSS was calculated by searching with the consensus motif sequences 5'-GNTANAAT and 5'-GG-N9-TCGT that we have identified respectively, along with the published sigma-28 motif 5'-CGATWT and distance parameters set by Porcelli *et al.* (2013). Two mismatches were allowed for the sigma-70 motif by following the criteria for published promoters. Of all promoter sequences upstream of identified Cappable-seq TSS, 4387 carried a sigma-70 motif, 51 contained a sigma-54 motif and 122 had a sigma-28 motif as visually presented in Figure 1d. Over half of all promoters for each sigma factor were internal and about a quarter of sigma-70 and sigma-28 promoters were antisense (Figure 1d). Interestingly, sigma-54 seems to have a higher proportion of primary promoters (33.33 %) compared to the other sigma factors.”

To address the reviewer's point, we have further analysed the expression levels between different sigma factors and TSS categories, and an additional figure was generated (please see: Figure S4). The following paragraph has been added to the main text.

“In order to explore the difference of TSS expression among all Cappable-seq TSS, the TSS expression coverage was calculated in transcripts per million (TPM) with hierarchical clustering of TSS categories and sigma motifs (Kröger *et al.*, 2013) (Figure S4.) The analysis revealed the most highly and widely expressed TSS across all conditions tend to be internal, likely due to the transcription signal from the protein-coding gene where the internal TSS resides. In addition, most antisense TSS have low expression in the majority of the conditions. However, condition-specific antisense TSS expression was observed, especially in the hyperosmotic stress (NaCl) condition. However, there was no clear expression variation observed between the TSS categories and no clustering of TSS with sigma motifs (Figure S4). Interestingly, most of the sigma-28 antisense TSS were on the opposite strand of metabolic pathways other than flagella-associated genes. This suggests that sigma-28 may regulate metabolic pathways other than flagellar assembly.”

“The difference of TSS positions among published results might result from different experimental methodologies (Dugar *et al.*, 2013; Porcelli *et al.*, 2013; Handley *et al.*, 2015). For instance, cells were harvested in different OD₆₀₀ values among the three studies. Moreover, Porcelli *et al.* (2013)'s samples were sequenced by the Roche 454 platform, while the data of Dugar *et al.* (2013) and Handley *et al.* (2015) were generated from the Illumina HiSeq platform.”

Response Figure 1: log₂TPM expression of all Cappable-seq. All TPM values were determined using the expression from the 10 bp upstream of each TSS. If a TSS has a TPM value of 0, its corresponding log₂TPM was transformed to 0. (Figure S4 in paper).

Parts of this manuscript were however difficult to understand with many of the results section lacking any introduction and assumptions of highly detailed knowledge of the methodologies. The most important of these aspects are listed under major changes and should be addressed prior to publication if this article is going to be widely accessible.

We apologise that some of the sections in the paper were difficult to follow. This is because the comprehensive nature of the study includes a wide range of data sets. We have now revised the paper thoroughly, re-written the complex sentences and expanded the introduction. In the result section, we have added introductory and explanatory sentences. We believe these changes would make the paper more accessible to a wider audience.

Examples of new introductory sentences are as follows:

“The novel TSS discovered by Cappable-seq may indicate the presence of novel sRNAs. In order to uncover more sRNAs, a list of sRNAs were predicted from several *in silico* tools (see method section)”

“Further, detailed investigation of the predicted sRNAs aimed to understand the conditions and pathways in which the sRNAs play important roles in stress adaptation. The biological function of the predicted sRNAs was investigated by identifying their potential binding targets, and assuming that the sRNA-target pairs share highly correlated gene expression patterns across all RNAtag-seq samples and have stable binding energy. An sRNA-target network was then built by analysing RNAtag-seq data with WGCNA, DESeq2 and IntaRNA, using similar criteria to a comparable study that constructed the RNA-RNA interactome from *Staphylococcus aureus* using RNA-seq datasets (Subramanian *et al.*, 2019) (Figure S9a).”

“Detailed investigation of individual analysis would also help to understand the conditions and pathways that play important roles in stress adaptation

major changes

The start of the results section describes results of TSS in previous studies. There is however no indication of species (presumably *C. jejuni*) or strain (all NCTC11168?) studied in these publications or what the environmental stresses were where this was examined. There also huge differences between the studies that are unexplained but presumably relate to methodology.

We apologise that the full description of the species was not mentioned at the beginning of the results section. We have now added the species name and strain (*C. jejuni* NCTC11168) at the beginning of section 3.1 and modified a paragraph in the introduction describing the previous studies mentioned in section 3.1, explaining the environmental stresses used.

We have made the following changes in the introduction:

“The lack of identified global RNA-binding proteins has hindered the discovery of *C. jejuni* sRNAs. There have been several published transcriptomic and RNA-Seq datasets (Dugar *et al.*, 2013; Porcelli *et al.*, 2013; Butcher and Stintzi, 2013; Taveirne *et al.*, 2013; Handley *et al.*, 2015) that have enabled the detection of potential novel sRNAs in *C. jejuni*. Of these, Dugar *et al.* (2013) and Porcelli *et al.* (2013) focused on standard growth conditions, whilst Butcher and Stintzi, (2013) investigated iron homeostasis and Handley *et al.* (2015) looked at inactivation of Fur and PerR. Notably, Taveirne *et al.* (2013) identified sRNA expression from an *in vivo* chicken model, although this was conducted in *C. jejuni* strain DRH212, which is a streptomycin resistant derivative of strain 81-176 rather than the more widely studied NCTC 11186 used in the aforementioned datasets. These studies have provided a beneficial insight into the transcriptional landscape of *C. jejuni*. However, they are not necessarily comparable due to differences in growth and stress conditions and may have missed condition-specific sRNAs outside the scope of their study. Bacterial sRNAs play a critical role in regulating stress responses as reviewed in Hoe *et al.* (2013), therefore a comprehensive approach is needed to uncover

the untapped reserve of potential condition-dependent novel sRNAs in order to fully understand the complex nature of *C. jejuni* post-transcriptional regulation. Among experimentally confirmed sRNAs, most of their condition-specific activities remain largely unknown.”

Changes in results section 3.1:

“Previous studies by Dugar *et al.* (2013) and Porcelli *et al.* (2013) have found TSS in *C. jejuni* NCTC11168 wildtype strain under standard laboratory growth conditions and Handley *et al.* (2015) in *C. jejuni* NCTC11168 *fur perR* mutant. These studies all used the dRNA-seq approach developed by Sharma *et al.* (2010). Dugar *et al.* (2013) identified 1905 TSS, with 1837 TSS retained after clustering TSS less than 10 bp apart. Porcelli *et al.* (2013) and Handley *et al.* (2015) reported 992 and 14 TSS respectively.”

Lines 196-199. These sentences are unclear with loose descriptions of the associations (nearby) and numbers (most, Some, several). These sentences I assume relate to the 116 sRNAs predicted by ANNOgesic but this is not entirely clear. Predictions of 5' ends should have some indications of distances so it is clear that they are reasonable.

We have elaborated on lines 196-199 in the original submission by adding clarification on how 5' ends were identified with our criteria.

The changes made in the paragraph are as follows:

“A closer inspection of the 5' and 3' boundaries of the ANNOgesic-predicted sRNAs revealed that those that matched with the benchmark sRNAs, had start sites with genomic positions less than 10 nucleotides away to the 5' ends of the benchmark sRNAs. For the ANNOgesic-predicted sRNAs that were associated with multiple TSS, the TSS furthest upstream was selected as the 5' end, unless gene expression coverage suggested otherwise. The 3'-ends of the ANNOgesic-predicted sRNAs, were further refined using similar parameters as toRNAado (<https://github.com/pavsaz/toRNAado>), which filtered out the 3'-end nucleotides with expression coverage at least 5-fold lower than the maximum expression of their corresponding predictions. Predictions that appeared to share similar genome coordinates with annotated genes were removed manually to minimise false positives from transcription signals of annotated genes. After manual correction and removal of predictions without TSS, 96 putative sRNAs were carried forward for downstream analysis.”

Lines 196-211. These paragraphs discuss the association of TSS with sRNA. Line 211 indicates that the TSS data was from all conditions which indicates that some sRNAs may have been missed by previous studies due to only using standard conditions. However the descriptions of the data do not give any indications of how many of the TSS were found under non-standard conditions and hence it is not

possible to properly assess if this is the reason that previous studies did not find them.

Some novel TSS were discovered due to the sensitivity of Cappable-seq over dRNA-seq. We have further explored TSS expression under non-standard conditions by including a new figure (Figure S8).

Moreover, the following explanation was added into the main text:

“Further analysis of the condition-specific expression of TSS showed that 3736 Cappable-seq TSS were found in the standard laboratory condition (37_M), while 4400 were found among the other 20 conditions. Only three TSS were expressed in 37_M but not in any other conditions (Figure S4). Likewise, among Cappable-seq TSS specifically associated with the 96 predicted sRNAs, 69 and 76 TSS were found in standard laboratory conditions (37_M) and non-standard conditions, respectively. All 69 TSS expressed in 37 M were also expressed in other conditions. Most predicted sRNAs carried TSS that showed varied levels of expression across non-standard conditions when compared to the standard 37 M condition (Figure S8). Interestingly, TSS expression of ANNOgesic-predicted sRNAs, such as CjSA21, CjSA54 and CjSA102, was higher in the hyperosmotic replicates (Figure S8) suggesting that some sRNAs might have stress regulatory roles. Note that some internal TSS may regulate the expression of antisense or UTR-derived sRNAs if the transcript boundaries of those sRNAs reach the antisense of UTR regions of protein-coding genes.”

Response Figure 2: \log_2 TPM expression of those Cappable-seq TSS that regulate ANNOgesic predicted sRNAs. All TPM values were determined using the expression from the 10 bp upstream of each TSS. If a TSS has a TPM value of 0, its corresponding \log_2 TPM was transformed to 0. (Figure S8 in paper)

Lines 227-228 refer to 11 expression modules. The data used for the derivation of these modules is not clear. Were these predictions made using the expression data from all of the different conditions or only some? If so wouldn't the modules vary depending on which conditions are used? Further, it is not possible to evaluate the nature of these modules as no data is provided on the numbers of genes in each module or indeed a list of the specific genes. Some indications of module content are provide but this is gleaned as the text progresses and is somewhat frustrating and hinders understanding. More detailed data on the modules will be required if other groups are going to generate comparative data or indeed to do further confirmatory analyses.

We would like to thank the reviewer for this very good suggestion. We have counted the number of genes in each cluster and generated a bar plot (Figure 3b), which has been added to the main text.

Response Figure 3: The number of genes assigned to co-expression modules I - XI. (Figure 3b in paper)

The description of the bar plot in the main text (Figure 3b) is as follows:

“Module X consisted of 653 genes, more than any other module. In contrast, module V was the smallest module that consisted of only 65 genes (Figure 3b).”

Furthermore, the following explanation on the co-expression analysis was included:

“Co-expression analysis was conducted on all annotated genes and predicted sRNAs. Based on the gene expression patterns across all RNAtag-seq conditions, we partitioned all genes into 11 co-expression modules using average linkage hierarchical clustering based on their expression patterns and network topologies.”

Minor

Line 38-39, colloquial use of lab and unclear use of transmission chain

This has been corrected. The relevant paragraph is as follows:

“While *C. jejuni* is a fastidious organism to culture in the laboratory, its ability to adapt to stress during transmission makes it a successful pathogen. *C. jejuni* experiences environmental stresses such as but not limited to bile salt, temperature variations, reactive oxygen species (ROS), and host iron limitation

(Flint *et al.*, 2012, 2014; Butcher *et al.*, 2015). It remains unclear how *C. jejuni* adapts to various environmental stresses with such a small genome (~ 1.6Mb) that carries only three annotated sigma factors and no conserved global stress response regulators like *rpoS* found in other Gram-negative bacteria (Parkhill *et al.*, 2000).”

Table 1. Could be more informative. Treatment could be split into two columns with initial growth conditions indicated in one column and incubations conditions in the second. Periods of incubation in specific conditions need to be indicated e.g. for 'acid' how long were cells subjected to incubation in MH2 broth at pH3.5.

We would like to thank the reviewer for this valuable suggestion. We have now revised Table 1.

Sample name	Initial growth	Treatment
37_M	exponential phase at 37 °C	NA
37_ES	early stationary phase at 37 °C	NA
37_LS	late stationary phase at 37 °C	NA
42_M	exponential phase at 42 °C	NA
42_ES	early stationary phase at 42 °C	NA

42_LS	late stationary phase at 42 °C	NA
cold	exponential phase at 37 °C	Resuspended in MH2 broth at 4 °C for 24 hours (Brown et al. , 2014).
5%_ce	exponential phase at 37 °C	Incubated in MH2 broth supplemented with 5% chicken exudate at 4 °C for 24 hours (Birk et al. , 2004)
acid	exponential phase at 37 °C	Resuspended in MH2 broth at pH 3.5 for 10 minutes (Le et al. , 2012)
ana	exponential phase at 37 °C	Incubated in anaerobic chamber for 1 hour
heat	exponential phase at 37 °C	Incubated at 55 °C for 3 minutes (Klančnik et al. , 2014).
iron_lim_M	exponential phase at 37 °C	Growth media was MEM– supplemented with 10 M pyruvate (Butcher and Stintzi, 2013).
iron_lim_E S	early stationary phase at 37 °C	Growth media was MEM– supplemented with 10 M pyruvate (Butcher and Stintzi, 2013).
iron_rep_M	early stationary phase at 37 °C	Growth media was MEM– supplemented with 10 M pyruvate and 40 M FeSO ₄ (Butcher and Stintzi, 2013).

iron_rep_ES	early stationary phase at 37 °C	Growth media was MEM– supplemented with 10 M pyruvate and 40 M FeSO ₄ (Butcher and Stintzi, 2013).
nacl	exponential phase at 37 °C	Incubated in 1.5% NaCl for 2 hours (Cameron et al. , 2012).
oxidative	exponential phase at 37 °C	Added 3mM H ₂ O ₂ for 10 minutes (Klanchnik et al. , 2006).
starv	early stationary phase at 37 °C	Resuspend in Ringer's solution for 5 hours (Klanchnik et al. , 2009).
GSNO	exponential phase at 37 °C	Incubate in 1.5 mM GSNO for 2 hours (Elvers et al. , 2005).
sod_deoxy_M	exponential phase at 37 °C	Growth media was supplemented with 0.1 % sodium deoxycholate (Malik-Kale et al. , 2008).
sod_deoxy_ES	early stationary phase at 37 °C	Growth media was supplemented with 0.1 % sodium deoxycholate (Malik-Kale et al. , 2008).

Response Table 1: All experimental conditions for Cappable-seq and RNAtag-seq. The cells were cultured in MH2 broth unless specified otherwise. (Table 1 in paper)

Figure 3. Legend: misspelling 'a person correlation'

Thank you for notifying us of the typo which has been fixed. "Pearson" is also capitalised later on in the paragraph. The corrected caption is seen below:

"The Pearson correlation between the eigengene (the first principal component of the gene expression matrix) of each co-expression module and experimental traits. All black boxes indicated a Pearson correlation value > 0.05 . A positive correlation suggested that under the corresponding experimental condition, the overall gene expression in the module increased as well. Likewise, a negative correlation indicated that the overall gene expression of the module decreased under that experimental condition."

Reviewer 2:

Li et al. present a comprehensive dataset of RNASeq analysis of *Campylobacter jejuni* (strain 11168) grown under 21 different in vitro conditions. The conditions selected are defined by the authors as "host and transmission relevant", and include those emphasizing temperature, acid, iron limitation, starvation, and many others. The major thrust of the work is aimed in particular at identifying transcription start sites and small non-coding RNAs that might contribute to fitness in these conditions. After some quality-control and filtering criteria were applied, 96 sRNAs were ultimately analyzed further. The sRNAs were categorized into 11 modules based on patterns of co-expression, which were further classified to some degree by KEGG pathway annotations within co-expression modules. One sRNA, CjSA21 was selected for further analysis, and putative targets for this were identified using the RNA:RNA interaction tool IntaRNA. This led the authors to suggest that CjSA21 might target the RNA for the chemotaxis motility transducer protein Tlp1-4, and co-expression of these RNAs occurred under 'food-processing' conditions (e.g., cold temperatures, presence of 'chicken exudate'), leading the authors to further speculate that this interaction might promote elevated biofilm growth (a sessile as opposed to motile lifestyle) during food processing.

This study represents a significant amount of work and valuable analysis of RNASeq data in an important foodborne pathogen. It provides a roadmap for others who might seek to do similar comprehensive analysis of sRNA function, and the experimental and bioinformatic approaches used by these authors will certainly be instructive to others. The major weakness comes from the highly speculative conclusions drawn by the authors based - for the most part - on co-expression and InstRNA findings, with very little experimental follow-up to test any specific hypotheses. While a weakness, this is not a major flaw in the work, but rather a statement of its limitations.

We would like to thank the reviewer for their positive feedback and for highlighting the limitations of our study. We believe the sRNA-mRNA interactions highlighted in this study to be interesting targets for future experimental follow-ups.

1. To be more rigorous in their scholarship, the authors must clarify their description of previous transcriptomic work on *C. jejuni*. They state in their abstract that "published *C. jejuni* sRNAs were discovered during growth under standard laboratory conditions..." Earlier work they cite (Taveirne et al) reported significant findings regarding sRNA expression from a transcriptomic analysis of *C. jejuni* 81176 harvested from experimentally-infected chickens. This needs to be clearly described in their rationale for this study, as the way they describe prior work in this area is misleadingly narrow.

Related to this point, the authors should compare the sRNAs they identified in this study to those described in the experiments from the chicken infection - to what degree do the two datasets (in vitro v. in vivo) overlap? The major comparisons carried out in this study are to RNASeq from other in vitro conditions.

Thank you for your feedback. We agree what was stated in the abstract conflicts with the paragraph in the introduction, so we have modified and added a sentence in the abstract to emphasise that other studies used specific conditions but many sRNAs may have been missed as they could be condition-dependent. We have added clarification and description of previously published transcriptomic studies of *C. jejuni* in the introduction and stressed that our rationale is to uncover more sRNAs by using a comprehensive approach rather than focusing on specific conditions.

Changes in the abstract:

"Published *C. jejuni* sRNAs have been discovered in specific conditions but with limited insights into their biological activities. Many more sRNAs are yet to be discovered as they may be condition-dependent."

Changes in the introduction:

"The lack of identified global RNA-binding proteins has hindered the discovery of *C. jejuni* sRNAs. There have been several published transcriptomic and RNA-Seq datasets (Dugar *et al.*, 2013; Porcelli *et al.*, 2013; Butcher and Stintzi, 2013; Taveirne *et al.*, 2013; Handley *et al.*, 2015) that have enabled the detection of potential novel sRNAs in *C. jejuni*. Of these, Dugar *et al.* (2013) and Porcelli *et al.* (2013) focused on standard growth conditions, whilst Butcher and Stintzi, (2013) investigated iron homeostasis and Handley *et al.* (2015) looked at inactivation of Fur and PerR. Notably, Taveirne *et al.* (2013) identified sRNA expression from an *in vivo* chicken model, although this was conducted in *C. jejuni* strain DRH212, which is a streptomycin resistant derivative of strain 81-176 rather than the more widely studied NCTC 11186 used in the aforementioned datasets. These studies have provided a beneficial insight into the transcriptional landscape of *C. jejuni*. However, they are not necessarily comparable due to differences in growth and stress conditions and may have missed condition-specific sRNAs outside the scope of their study. Bacterial sRNAs play a critical role in regulating stress responses as reviewed in Hoe *et al.* (2013), therefore a comprehensive approach is needed to uncover

the untapped reserve of potential condition-dependent novel sRNAs in order to fully understand the complex nature of *C. jejuni* post-transcriptional regulation. Among experimentally confirmed sRNAs, most of their condition-specific activities remain largely unknown.”

With regards to the reviewer’s second point, as the study by Taveirne *et al.* (2013) was carried out in another strain, we do not think it is appropriate to compare our dataset to theirs. BLASTn analysis of *C. jejuni* ncRNAs suggests that sequence conservation is limited in this species (Dugar *et al.*, 2013).

2. A significant amount of the data in this paper is reported in supplemental files and that should be reconsidered. Figures S8 and S11, for examples, should be included along with the main figures and not in supplemental material.

Both supplementary figures have now been included in Figure 3. The updated figure is seen below:

Response Figure 4: (a) PCA plot of transformed raw expression data from all 21 conditions. (b) The number of genes assigned to co-expression modules I - XI (c) The Pearson correlation between the eigengene (the first principal component of the gene expression matrix) of each co-expression module and experimental traits. All black boxes indicated a Pearson correlation p -value > 0.05 . A positive correlation suggested that under the corresponding experimental condition, the overall gene expression in the module increased as well. Likewise, a negative correlation indicated that the overall gene expression of the module decreased under that experimental condition. (d) $-\log_{10}(\text{FDR})$ of KEGG pathways enrichment for all co-expression modules. Those with $\text{FDR} > 0.05$ are coloured in white. Figure 3 in paper)

Other changes:

- All in-text citations changed to correct format.

- Changed all instances of “TSSs” to “TSS”
- Line 73 - changed “Magnetic Beads” to “enrichment”
- Line 147 - changed “Functional” to “functional”
- Line 173 - changed “in” to “identified from”
- Line 182 - changed “factors” to “factor”
- Line 206 - removed “the”
- Line 235 - added “contribution from the”
- Line 241 - changed “typical” to “common”
- Line 264 - added “data”
- Line 277-278 - added “and may involve crosstalks with other stress response pathways.”
- Line 285 - Modified sentence to “Meanwhile, modules II, X and XI consisted of 274, 653 and 125 genes, respectively. They were also enriched with over 10 KEGG pathways.”
- Line 284 - changed “showed the over representation” to “enriched with”.
- Modified section 3.3.2 to “DESeq analysis identified differentially upregulated and downregulated genes under specific conditions. Pathways involved in general metabolism such as “Biosynthesis of amino acids”, “ABC transporters” and “Two-component system” were significantly enriched at least 24 out of 25 pairwise comparisons (Figure 11a). The genes in these pathways were likely to play a more prominent role in stress adaptation. In contrast, some pathways demonstrated differential expression in very few pairwise comparisons. These pathways included “beta-Lactam resistance” and “Tyrosine metabolism” that were enriched in only two and one pairwise comparison, respectively (Figure 11b). Hence, these pathways may be less affected by stress conditions”
- Line 303 - removed “functional”, added “The enriched pathways were”
- Line 306 - modified sentence to “After combining the results from WGCNA, DESeq2 and IntaRNA, 513 sRNA-target interactions (Figure S12) fulfilled all three criteria for our RNA-RNA interactions (described in Figure S9a legend).”
- Line 310 - added “sRNA with targets over represented in at least one pathway are shown in”
- Line 319 - changed “was” to “is” and added “and is potentially”
- Line 321 - added “also”
- Line 329 - added “is” and “of”
- Line 330 - added “is”
- Line 332 - modified sentence to “Both “Two-component system” and “Bacterial chemotaxis” were also among the differentially enriched pathways that appeared in 24 and 17 out of 25 pairwise comparisons respectively”
- Line 335 - modified paragraph to “Enrichment analysis of module II genes showed that the most significantly enriched KEGG pathways were “Bacterial chemotaxis” and “Two-component system”, as module II includes genes such as *tlp1-4*, *cheA* and *cheV*. Notably, module II was also enriched with amino acids and fatty acids biosynthetic pathways (Figure 4b and S14). A comparison of variance stabilizing transformation (vst) expression between CjSA21 and *tlp1-4* demonstrated statistically significant bicor coefficients between -0.567 and -0.794 (Figure 4c).”
- Table 3 legend - corrected leq to \leq
- Line 375 - added “different by”, removed “off”

- Line 379 - modified sentence to “CjSA21 may regulate the translation or stability of *t/p1-4* without affecting its transcript abundance.”
- Line 399 - changed “bolded” to “highlighted in bold”
- Line 417 - added “are also”
- Line 425 - corrected OD⁶⁰⁰ to OD₆₀₀

May 10, 2022

Dr. Chrystala Constantinidou
University of Warwick
Microbiology and Infection Unit
Warwick
United Kingdom

Re: Spectrum00203-22R1 (Post-transcriptional regulation in response to different environmental stresses in *Campylobacter jejuni*)

Dear Dr. Chrystala Constantinidou:

Your manuscript has been accepted, and I am forwarding it to the ASM Journals Department for publication. You will be notified when your proofs are ready to be viewed.

Sincerely,

S. Wesley Long
Editor, Microbiology Spectrum

Journals Department
Supplemental Material FOR Publication: Accept